# Discharge Interventions for First Nations People with Injury or Chronic Conditions: A Protocol for a Systematic Review

**DOI:** 10.3390/ijerph191811301

**Published:** 2022-09-08

**Authors:** Julieann Coombes, Andrew J. A. Holland, Kate Hunter, Keziah Bennett-Brook, Courtney Ryder, Summer M. Finlay, Phillip Orcher, Mick Scarcella, Karl Briscoe, Dale Forbes, Madeleine Jacques, Roland Wilson, Elizabeth Bourke, Camila Kairuz

**Affiliations:** 1The George Institute for Global Health, Sydney, NSW 2042, Australia; 2Faculty of Medicine, University of New South Wales, Sydney, NSW 2052, Australia; 3Department of Paediatric Surgery, The Children’s Hospital at Westmead, Westmead, NSW 2145, Australia; 4Faculty of Medicine and Health, The University of Sydney, Sydney, NSW 2006, Australia; 5Indigenous Health College of Medicine and Public Health, Flinders University, Adelaide, SA 5042, Australia; 6School of Population Health, University of New South Wales (UNSW), Sydney, NSW 2052, Australia; 7School of Health and Society, Wollongong University, Wollongong, NSW 2522, Australia; 8Agency for Clinical Innovation, Sydney, NSW 2065, Australia; 9The Sydney Children’s Hospital Network (SCHN), Sydney, NSW 2145, Australia; 10National Association of Aboriginal and Torres Strait Islander Health Workers and Practitioners (NAATSIHWP), Canberra, ACT 2606, Australia; 11Department of Community and Justice, Sydney, NSW 2012, Australia

**Keywords:** aftercare, injury, chronic conditions, discharge plan, first nations, systematic review

## Abstract

Severe injury and chronic conditions require long-term management by multidisciplinary teams. Appropriate discharge planning ensures ongoing care to mitigate the long-term impact of injuries and chronic conditions. However, First Nations peoples in Australia face ongoing barriers to aftercare. This systematic review will locate and analyse global evidence of discharge interventions that have been implemented to improve aftercare and enhance health outcomes among First Nations people with an injury or chronic condition. A systematic search will be conducted using five databases, Google, and Google scholar. Global studies published in English will be included. We will analyse aftercare interventions implemented and the health outcomes associated. Two independent reviewers will screen and select studies and then extract and analyse the data. Quality appraisal of the included studies will be conducted using the Mixed Methods Appraisal Tool and the CONSIDER statement. The proposed study will analyse global evidence on discharge interventions that have been implemented for First Nations people with an injury or chronic conditions and their associated health outcomes. Our findings will guide healthcare quality improvement to ensure Aboriginal and Torres Strait Islander peoples have ongoing access to culturally safe aftercare services.

## 1. Introduction

The complex nature of chronic conditions and severe injuries require multidisciplinary management over extended time periods and monitorisation to promote optimal rehabilitation when possible and improve overall health and wellbeing [1]. Thus, patients admitted to hospital with severe injury or chronic conditions often require ongoing health care after hospital discharge.

Given the growing number of people suffering from chronic conditions, healthcare systems that have been traditionally designed for acute illness face increasing pressure to meet the needs for the ongoing support of patients with complex and chronic conditions [2]. This has led to increasing research on system changes and strategies to improve chronic illness care, including increased use of community resources, telehealth, self-management support and coordinated care [3]. Discharge planning is currently considered paramount to guaranteeing continuity of care for patients with chronic or complex conditions [4,5,6,7,8,9]. Studies have reported multiple benefits of implementing discharge planning, such as satisfaction with discharge, reduction in the hospitalisation length of stay and hospital readmissions within 30 days of discharge [5].

Due to ongoing colonisation, Aboriginal and Torres Strait Islander people’s hospitalisation rate is 2.3 times the rate of non-Indigenous Australians [10]. According to the study conducted by Al-Yama, Aboriginal and Torres Strait Islander people experience 284 years lost per 1000 people due to premature death or living with disease [11]. Chronic diseases, including injury, are estimated to cause more than half (64%) of the disease burden [11]. Despite solid evidence on the importance of ensuring ongoing aftercare and discharge planning for people with chronic conditions, research has found that Aboriginal and Torres Strait Islander peoples encounter significant barriers to aftercare, including miscommunication, lack of cultural safety, distance to medical treatment, and racism [12,13].

When designing and delivering healthcare services to First Nations people, special considerations need to be taken to ensure that those services are culturally safe and effective [14]. Research has evidenced that unconscious bias held by the dominant Western medical system is a significant factor contributing to worldwide racial disparities in quality health care [15]. In Australia, ongoing experiences of collective, historical, and transgenerational trauma due to colonialism have generated distrust and suspicious attitudes among Aboriginal and Torres Strait Islander people towards the medical system and services [16,17]. In addition, healthcare delivery is currently based only on a Western Biomedical model of health that fails to consider and therefore meet Aboriginal and Torres Strait Islander worldviews and health and wellbeing paradigms.

The national government and other organisations have recognised the need to provide healthcare services that are tailored to the cultural needs of Aboriginal and Torres Strait Islander people to move forward to safe, accessible, and culturally responsive health systems [18,19,20,21]. In this sense, it is important to consider cultural differences when designing and implementing quality improvement strategies to ensure ongoing aftercare for Aboriginal and Torres Strait Islander patients.

First Nations peoples in colonised countries are culturally very diverse and unique. Despite this, they all share a history of trauma, dispossession, cultural genocide, and experience culturally oppressive and racist healthcare systems that have failed to meet their cultural needs [22]. This is reflected in the inequitable health outcomes that First Nations peoples experience globally, even within low-income countries where non-Indigenous populations also experience high disease burden and mortality [22,23]. For instance, First Nations peoples in New Zealand, Canada, and the United States experience a gap in life expectancy of 8.5, 5–7 and 5 years, respectively [23,24]. Further, First Nations peoples worldwide are experiencing an increasingly disproportionate burden of chronic conditions as urbanisation continues to increase [25]. This has prompted a call for global development and provision of health services that are culturally appropriate to reduce the health gap [22,26].

For this reason, learning from culturally safe and successful interventions to improve healthcare delivery for First Nations communities globally can guide the implementation of strategies to improve healthcare delivery for Aboriginal and Torres Strait Islander people in Australia. Thus, the objective of the proposed systematic review is to locate and analyse discharge interventions that have been implemented globally to optimise aftercare care for First Nations peoples with injury or chronic conditions. This will inform the design and implementation of interventions to ensure Aboriginal, and Torres Strait Islander people with injury and chronic conditions have ongoing access to culturally safe care.

## 2. Materials and Methods

This protocol was prospectively registered in PROSPERO (ID CRD42021254718). A PRISMA-P Checklist for Systematic Review Protocols [27] has been completed and is available as Appendix A. We will follow the reporting guidelines and criteria set in the Updating guidance for reporting systematic reviews (PRISMA 2020) [28].

Additionally, we will follow the guidelines from the Australian Institute of Aboriginal and Torres Strait Islander Studies for ethical research in Indigenous studies [29], the guidelines for ethical conduct in Aboriginal and Torres Strait Islander health research (National Health and Medical Research Council, 2018) [30] and the Lowitja’s institute practical guide for researching Indigenous health [31].

### 2.1. Research Questions

What discharge interventions have been implemented and evaluated globally for First Nations people with an injury or a chronic condition?What effects on health outcomes have been found with the existing interventions?

### 2.2. Search Strategy

To answer the research questions, a systematic search will be conducted for published and grey literature using the following databases: PubMed, CINAHL, ProQuest, Embase, Web of Science, Google, and Google scholar. We developed a search strategy using the Boolean operators “AND” and “OR” and a combination of key terms related to “First Nations”, injury”, “chronic conditions”, and “discharge intervention”. A list of the key terms used to develop the search strategy can be seen below in Table 1. The search terms will be adapted to each database. The search strategy planned for each database is available in Appendix A.

To locate grey literature, we will review the first 10 pages of Google results. The results of all databases and google searches will be exported to the reference manager EndNote and duplicates will be removed. All titles and abstracts will be screened by two independent reviewers to select eligible studies. Then, the full text of preselected studies will be assessed by two independent reviewers against the following criteria:

### 2.3. Inclusion Criteria

Studies published in English between 1 January 2000 and 1 July 2022.Studies that included First Nations people of all ages with an injury or a chronic condition.Studies conducted in any country.Evaluated the implementation of a discharge plan or any discharge intervention in any healthcare setting.Studies reporting health outcomes and/or patient satisfaction associated with the studied intervention.Studies reporting other outcomes associated with the intervention such as Emergency Department presentations, hospital re-admissions and medical appointment attendance.Primary studies using mixed methods or qualitative or quantitative methods [case-control, cross-sectional, cohort, randomised controlled trials (RCTs) and controlled clinical trials (CCTs)].

### 2.4. Exclusion Criteria

Studies conducted on non-indigenous people.Published in a language different from English.Evaluated an intervention other than a discharge plan or related to patient discharge.Didn’t evaluate any discharge plan or discharge-related intervention.

Any discrepancies between the reviewers during the screening process will be resolved by dialogue until consensus is reached or with the assistance of a third reviewer if needed. The results of the screening and study selection process will be outlined in a PRISMA flow diagram.

### 2.5. Data Extraction

Once the study selection process has been completed, one team member will extract the following data when available and will organise it in an excel spreadsheet: basic study information (title, year of publication, country, authors, type of document, and journal) population and setting (sample size, sample source, age range, health condition, ethnicity, health setting) study methods (objectives, study design, quantitative or qualitative methods used, evaluated outcomes) and results (resulting themes, and associations with health outcomes). Another reviewer will check all data introduced to the spreadsheet by comparing it to the data reported in each paper to avoid possible errors.

### 2.6. Data Analysis and Synthesis

Studies meeting eligibility criteria will be analysed independently by two reviewers who will apply a decolonising lens. Through a decolonising lens, research discourse is shifted from a deficit discourse that portrays First Nations peoples as defective individuals that need to be fixed to a strength-based approach. In addition, by applying a decolonising lens, researchers reflect on the impact of colonisation upon First Nations peoples and the power imbalances perpetuated by a Western and oppressive system that privileges only the voices of those who are part of this dominant system [32]. After independent analysis, both reviewers will discuss and agree on a final data analysis. Any discrepancies between the reviewers at any stage of the data extraction and analysis will be resolved through discussion until consensus is reached or with the help of a third reviewer if needed. Findings will be presented through a narrative descriptive synthesis of quantitative findings, whilst an inductive thematic analysis will be performed for qualitative studies [33].

### 2.7. Quality Assessment

Two independent reviewers will use the Mixed Methods Appraisal Tool (MMAT) [34] to assess the quality of the studies included. The MMAT is a comprehensive tool for reviews that include qualitative, quantitative, and mixed methods studies and meets the accepted standards for validity and reliability [35,36]. Both reviewers will compare and discuss the results, which will be portrayed in tables.

In addition, we would like to highlight the importance of considering the impact that colonisation has brought upon Frist Nation peoples in different aspects, including research practices. For this reason, papers included in this systematic review will be appraised on the extent to which research complies with the criteria included in the CONSIDER statement [37]. The CONSIDER statement is a checklist including eight domains and 17 criteria for the reporting of research involving First Nations peoples. It was designed to strengthen the reporting of health research involving First Nations peoples and promote research approaches that are underpinned by First Nations participation, knowledge, and priorities to advance Indigenous health outcomes [37].

## 3. Discussion

We presented the protocol for a systematic review of global studies analysing discharge interventions implemented among First Nations people of all ages with an injury or chronic condition. To the best of the author’s knowledge, this is the first protocol for a systematic review that will analyse discharge interventions among First Nations peoples. The findings of the review will be analysed and compared with the available literature, and conclusions will be drawn highlighting strategies to provide better aftercare. Through the analysis of the studies that will be found in this review, we will acquire knowledge to inform healthcare quality improvements so as to ensure Aboriginal and Torres Strait Islander peoples have ongoing access to culturally safe aftercare services.

## 4. Conclusions

The findings of this systematic review will give clear guidelines of the elements that constitute culturally safe interventions to ensure ongoing care for First Nations people with injury or chronic conditions. These findings will inform a unique program to ensure ongoing burn care for First Nations children in Australia. It will also be of use for practitioners, researchers, First Nations communities, policymakers and other stakeholders interested in achieving better health outcomes for First Nations people with injuries and chronic conditions.

## Figures and Tables

**Table 1 ijerph-19-11301-t001:** Key terms.

First Nation Peoples	Chronic Conditions	Discharge Intervention
Indige * Aborigin * indian Torres Strait Islander First People Inuit First Nation Māori Native American Native Sami Indians	Injur * Burns Chronic condition * Heart disease * Chronic kidney disease Cystic fibrosis Diabetes COPD Chronic obstructive pulmonary disease Chronic illness Chronic disease Chronically ill Long term conditions	Discharge plan * discharge Patient discharge Discharge process Discharge management Discharge education Discharge intervention Follow-up Aftercare Model of aftercare

Note: The asterisk (*) represents any group of characters, including no character used in some databases to broaden the search.

## Data Availability

No applicable.

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
