# Peer review of "Discharge Interventions for First Nations People with Injury or Chronic Conditions: A Protocol for a Systematic Review"

_ijerph, 2022, doi:10.3390/ijerph191811301_

Round 1

Reviewer 1 Report

Due Aug 30, 2022

Review request: International Journal of Environmental Research and Public Health

Manuscript type: Protocol

Manuscript ID: ijerph-1875834

Title: Discharge interventions for First Nations people with injury or chronic conditions: A protocol for a systematic review

Synopsis

This manuscript is written about the registered protocol in  PROSPERO (ID CRD42021254718) for a systematic review study. The aim of the protocol is to investigate the association between the hospital’s aftercare interventions implemented and health outcomes in First Nations people with injury or chronic conditions. First Nations people are communities of ancestrally indigenous or aboriginal people who were dwelling long before colonization. The colonizers who overtook these First Nations people have established the healthcare system currently in place. The following guidelines were utilized for scientific validity:

·         Mixed methods appraisal tool (MMAT, Hong, 2018): McGill U Canada, National Collaborating Centre for Methods and Tools. “systematic mixed-studies reviews” The tool outlines a set of criteria and screening questions to provide an overall quality score.

·         CONSIDER Statement (Mayer, 2021): Consolidated Recommendations for Sharing Individual Participant Data from Human Clinical Studies

·         PRISMA (Liberati, 2009) Preferred Reporting Items for Systematic Reviews and Meta-Analyzes has its origin in QUOROM (Quality of Reporting of Meta-Analyses) of studies that evaluate healthcare interventions

·         PRISMA-P (Shamseer, 2015) 

·         Two supplements documents: PRISMA-P 2015 Checklist and Search terms of databases PubMed, CINAHL, ProQuest, Embase, Web of Science, Google, and Google Scholar.

All are current and well accepted guidelines.

Reviewer's conflict of interest: None

Comments and other specific recommendations are listed below:

1.       Title.

a.       Remove a period at the end of the title.

b.       The use of the term “First Nations people” needs to be examined from the context of the manuscript. Aboriginal, indigenous, native, etc., are more commonly used terms in other parts of the world. The fact that all authors are based in Australia may imply the external validity of this study is limited to the aboriginal and Torres Strait Islander people; however, the word “global” in the manuscript as well as the search methodology implicate that the external validity of health inequity in other indigenous populations is not limited to the Torres Strait Islander people.

2.       Abstract.

a.       Line 20, “First Nations peoples in Australia face ongoing barriers to aftercare.” Provide a number (statistics) such as the number used in the introduction section, “Aboriginal and Torres Strait Islander peoples’ hospitalization rate (overall hospitalization rate or readmission after discharge?) is 2.3 times the rate of non-Indigenous Australians.” (Line 51)

b.       The above statement also indicates that this research question (#1) is prompted by the ongoing health inequity in the Australian healthcare system, in which the authors attempt to draw ‘global’ conclusions by this systematic review. I recommend that authors let readers know what prompted this study and how the study is needed globally by citing statistics of other countries such as the Native Americans in the U.S.   

3.       Introduction.

a.       The introduction is well presented to demonstrate the needs leading to the research question; however, citing the statistics of other countries may strengthen the needs statement for a global study.

b.       How do the authors consider the colonization of African nations or South American nations by Belgium, Germany, Portugal, France, and the Netherlands, for example.

4.       Materials and Methods.

a.       The protocol was registered in the PROSPERO and the PRISMA-P Checklist has been completed. Culturally appropriate ethical approvals were sought. A plan for using a PRISMA flow chart was indicated. They were all current and well prepared.

b.       For Table 1, a caption should explain the meaning of the asterisk (*).

c.       The exclusion criterion states “Studies conducted on non-indigenous people.” Is the comparative study included?

d.       The definition of a ‘decolonising lens’ is described as “research discourse is shifted from a deficit discourse that portrays First Nations peoples as defective individuals that need to be fixed to a strength-based approach.” (Line 156) The qualitative perspective is well explained.

5.       Results. Results do not exist.

6.       Discussion.

a.       A limitation may include the development of adaptive (or maladaptive) healthcare policy to culturally appropriate care of indigenous populations in different parts of the world. If the discussion is limited to the Australian First Nations people, readers may misunderstand the intentions of the global findings.

b.       In order to clarify the distinction between the global findings and the Australian First Nations peoples’ findings, the research question #1 can be separated to: 1(a) and 1(b) as the 1(a) will be the mixed method findings for the Australian First Nations people and 1(b) will be the mixed method finding for the global indigenous communities.

7.       Conclusion. No comment.   

End of review.

Reviewer 2 Report

Thank you for giving me the opportunity to read this protocol. The fact that it is already published in Prospero supports its suitability, however I have some recommendations:

-I recommend not including Google as a search tool

-I recommend expanding the search language to French, since it is expected that they will find very interesting local literature (Canada).

-I do not understand why the inclusion criteria indicate "Studies conducted in any country" (are there people from the first nations in any country?), nor why the search is limited to the year 2000.

-In the discussion they mention a very specific topic that is completely out of context: "In addition, findings from this review will guide an intervention to provide culturally safe ongoing burn care and a clear pathway for Australian First Nations children with burn injuries." I think this is another specific objective that should be answered with a different review.

-I think they will have serious difficulties comparing such different health contexts (Australia, New Zealand, USA, Canada...) and that it would be convenient for them to focus their search and objectives taking these circumstances into account (public centers, private centers... .).

Good luck with this nice and interesting revision.
